# Preparation of Hot-Melt Extruded Dosage Form for Enhancing Drugs Absorption Based on Computational Simulation

**DOI:** 10.3390/pharmaceutics12080757

**Published:** 2020-08-11

**Authors:** Sung-Min Choi, Sung-Hoon Lee, Chin-Yang Kang, Jun-Bom Park

**Affiliations:** 1College of Pharmacy, Sahmyook University, Seoul 01795, Korea; senttome@naver.com (S.-M.C.); sunghoon@cju.ac.kr (S.-H.L.); kangjy@syu.ac.kr (C.-Y.K.); 2Department of Pharmaceutical Engineering, Cheongju University, Cheongju 363-763, Korea; 3Bioavailability Control Laboratory, Sahmyook University, Seoul 01795, Korea

**Keywords:** hot-melt extrusion technology, cilostazol, dissolution rate and permeability, PBPK simulation, parameter-sensitive analysis

## Abstract

The aim of this study was to control the dissolution rate and permeability of cilostazol. To enhance the dissolution rate of the active pharmaceutical ingredient (API), hot-melt extrusion (HME) technology was applied to prepare a solid dispersion (SD). To control permeability in the gastrointestinal tract regardless of food intake, the HME process was optimized based on physiologically based pharmacokinetic (PBPK) simulation. The extrudates were produced using a laboratory-scale twin-screw hot-melt extruder with co-rotatory screws and a constant feeding rate. Next, for PBPK simulation, parameter-sensitive analysis (PSA) was conducted to determine the optimization approach direction. As demonstrated by the dissolution test, the solubility of extrudate was enhanced comparing cilostazol alone. Based on the PSA analysis, the surfactant induction was a crucial factor in cilostazol absorption; thus, an extrudate with an even distribution of lipids was produced using hot-melt extrusion technology, for inducing the bile salts in the gastrointestinal tract. In vivo experiments with rats demonstrated that the optimized hot-melt extruded formulation was absorbed more rapidly with lower deviation and regardless of the meal consumed when compared to marketed cilostazol formulations.

## 1. Introduction

It is well known that solubility and permeability of a drug in the gastrointestinal tract is very crucial to exert its effect following oral administration. The Biopharmaceutical Classification System (BCS) was introduced to classify drug substances according to their aqueous solubility and intestinal permeability [1]. Currently, numerous studies on the solubilization of poorly water-soluble drugs are ongoing. The proposed solubilization methods include the preparation of salts, water-soluble prodrugs, use of surfactants, or the formation of solid dispersions (SDs) [2,3].

In this study, hot melt extrusion (HME) technology was applied to prepare SD, because it does not pose any toxicity problems associated with residual solvent. Thus, this technology can be called environmentally friendly [4,5,6]. Furthermore, HME is a well-known manufacturing technique used to enhance the dissolution rate of poorly water-soluble drugs, converting drugs into their amorphous forms at high temperature and pressure, and evenly dispersing the active pharmaceutical ingredient (API) into the hydrophilic polymer [7,8,9,10].

However, enhancing or controlling a drug’s permeability in our body is not as easily achieved compared to controlling dissolution rate. Notably, several drugs currently under development or marketed are poorly water-soluble or have a unique absorptive factor, leading to limited absorption following oral administration [11,12]. To control the permeability of drugs, an understanding of physiologically based pharmacokinetic (PBPK) is essential. However, in vivo pharmacokinetic (PK) observations and the measurement of absorption are extremely time-consuming and expensive processes, along with concerned animal ethical issues. Thus, PBPK modeling based on computer simulations is now gaining focus. In addition to drug development, PBPK modeling has now become a necessity to solve these problems [13,14]. Indeed, more than 250 pharmaceutical companies have been using a simulation program based on PBPK modeling [12]. The development of PBPK modeling using computational simulation programs enables the advance prediction of actual in vivo PK and absorption measurements [15]. Furthermore, it reduces the cost and time spent by pharmaceutical companies performing in vivo and in vitro experiments [13]. The computational simulation program has been developed to a very high level, allowing the consideration of all routes of administration, absorption carriers, and individual enzyme specificity [16,17,18,19].

The aim of this study was to control the dissolution rate and permeability of cilostazol. Cilostazol (32.4 mg/mL of water solubility and 3.38 of log P) was selected as a model drug owing to its poor water [20]. It is also known to have a profound effect on drug absorption depending on whether food is consumed [21]. To enhance the dissolution rate of the API, hot-melt extrusion (HME) technology was used to prepare a SD. Additionally, in order to control the gastrointestinal permeability, the HME formulation was optimized based on PBPK simulation results. We investigated the oral pharmacokinetics of HME formulation and Pletaal^®^ (marketed cilostazol) in rats and observed the effects in food intake.

## 2. Materials and Methods

### 2.1. Materials

Cilostazol (USP grade) was gifted from BioChem (Seoul, Korea). Vinylpyrrolidone-vinyl acetate copolymer (Kollidon^®^ VA64), polyvinyl caprolactam–polyvinyl acetate–polyethylene glycol graft copolymer (Soluplus^®^), d-α-Tocopheryl polyethylene glycol 1000 succinate (TPGS^®^), and Polyethoxylated castor oil (Cremophor^®^) were kindly donated by BASF (Ludwigshafen, Germany); and microcrystalline cellulose (Avicel^®^) was provided by FMC Korea (Seoul, Korea). High-performance liquid chromatography (HPLC)-grade methanol (MeOH) and acetonitrile (ACN) were purchased from Burdick & Jackson (Muskegon, MI, USA). All other chemicals and solvents used were of analytical grade or above. The Gastroplus™ software for PBPK simulation was purchased from SimulationsPlus (San Diego, CA, USA), and the basic database and physiological data provided by Gastroplus™ was used for drug simulation. Pletaal^®^ tablets were purchased from Otsuka Pharmaceutical (Tokyo, Japan).

### 2.2. Preparation of Hot-Melt Extrudates

Three types of extrudates were produced using a laboratory-scale twin-screw hot-melt extruder, with co-rotatory screws and a constant feeding rate (11 mm Process 11^TM^, ThermoFisher Scientific, Dieselstraße, Karlsruhe, Germany). The first extrudate was prepared based on the method of Chen et al., which is a hot-melt granulation (HMG) method with cilostazol [22]. In case of the second, hot melt extrusion (HME) process used, Soluplus^®^ was used as the solubilizer instead of Avicel^®^ and the drug-loading was increased to 40%.

The last, HME process also used was modified based on the computational simulation approach. Kollidon^®^ VA64 (20%) was selected as a primary matrix former; TPGS^®^ (32%) and Cremophor^®^ (8%) were used as solubility enhancers. Detailed formulation compositions and processing conditions are presented in Table 1.

The hot melt granules (1st formulation) or hot melt extrudates (2nd and 3rd) with a size of 100–200 um were selected and applied to all in vitro and in vivo experiments.

### 2.3. Drug Release and Content Uniformity

The release behavior of pure cilostazol and its extrudates was evaluated using the dissolution test. The dissolution tests were performed using a PTWS-121C^TM^ 12 bowls dissolution tester with the USP apparatus II (37.5 ± 0.5 °C) at a paddle speed of 75 rpm. One liter of distilled water containing 0%, 0.1%, 0.3%, and 0.5% sodium lauryl sulfate (SLS) were used as the dissolution media. These media were developed according to the USP 42-NF37 “Cilostazol Tablets” [23]. Next, 5 mL of the dissolution fluid was collected, and the samples were diluted with ACN and MeOH (water 50%: ACN 35%: MeOH 15%) for HPLC analysis with a flow rate of 1 mL/min. The diluted sample was analyzed using the Agilent 1200 series HPLC system. A Fortis C18 column was used with a UV detector at a wavelength of 257 nm. Dissolution rate (%) was then calculated as follows.
(1)$$\mathrm{Dissolution}\;\mathrm{rate}\;\left(\%\right)=\frac{\mathrm{Absorbance}\;\mathrm{of}\;\mathrm{the}\;\mathrm{test}\;\mathrm{solution}\;\times \;\mathrm{concentration}\;\mathrm{of}\;\mathrm{the}\;\mathrm{Standard}\;\mathrm{solution}\;\left(\mathrm{mg}/\mathrm{mL}\right)\times \mathrm{Volume}\;\mathrm{of}\;\mathrm{Medium}\;}{\mathrm{Absorbance}\;\mathrm{of}\;\mathrm{standard}\;\mathrm{solution}\;\times \;\mathrm{Theoretical}\;\mathrm{Cilostazol}\;\mathrm{amounts}\;}\times 100$$

The drug content in the formulations were also evaluated. A standard solution containing approximately 10 mg of cilostazol in 100 mL of mobile phase (water 50%:ACN 35%:MeOH 15%) was prepared and dissolved using an Ultrasonic Ben 5510 DTH sonicator (Branson, Danbury, CT, USA) for 10 min. The test samples were prepared at the same concentration as cilostazol in the standard solution. Next, the samples were filtered through a 0.45 µm syringe filter (MiliporeSigma, Germany). After removing gas from the solution by using a sonicator, the standard and test samples were analyzed using a HPLC system. Three types of formulations were evaluated over 6 times.

The linearity of the calibration curve with above HPLC method was evaluated at six concentrations of cilostazol, ranging from 20 to 120 µg/mL (Appendix A). The regression equation was “y = 2607.1x − 0.7655; R^2^ = 0.9996”. The calibration curves were reproducible, and the relative standard deviations of all concentrations were within ±2.0%.

### 2.4. Computational PBPK Simulation

In the in vivo experiment, six rats weighing approximately 300 g were used. Hence, in the simulation using Gastroplus™, PBPK modeling assumed the rat weight as 300 g. The physiological parameters of 300 g rats provided by Gastroplus™ are shown in Table 2 and a parameter-sensitive analysis (PSA) using Gastroplus™ (v9, Simulations Plus, Lancaster, CA, USA) as a PBPK simulation was conducted to determine the direction in which cilostazol was to be formulated. For the PSA, the drug particle density was increased from 0.12 to 12 g/mL, in multiples of three. The precipitation particle radius was also increased from 0.2 to 20.0 μm, in multiples of three [24]. The bile salt solubilization ratio was calculated by comparing the difference between the lowest and highest values and increasing the values by three times, from 4.7 × 10^2^ to 4.7 × 10^4^.

### 2.5. Physicochemical Properties

Raman spectroscopy, X-ray diffraction (XRD), and scanning electron microscopy (SEM) were performed to determine the physicochemical properties of the hot-melt extrudate, which was processed based on computational simulation. Raman analysis was performed using a portable Raman spectrometer (BWTEK BWS465-532S, Newark, NY, USA). The analytical time was 10 s, and the laser was injected five times. For XRD, the Bruker D8 Focus (Bruker AXS GmbH, Karlsruhe, Germany) at 40 kV and 50 mA was used. The XRD data were collected in increments of 0.02° and the 2 range from 10–50° at 1 s/step. SEM was used to detect any physical changes in the matrix systems. The images were collected using a JSM-6510 scanning electron microscope (JEOL Ltd., Tokyo, Japan) at 20 kV and magnification of 500 ×. To increase sensitivity, before sample detection, samples were coated with gold nanoparticles. The images were acquired under the same conditions.

### 2.6. In Vivo Test

The actual mean weight of the six rats used the in vivo test was 304 ± 2.87 g. The rats were divided into the experimental and control groups, comprising of three rats each. The experimental group was administered the third hot-melt extrudates (computational simulation-based formulation); Pletaal^®^ was administered to the control group. Both groups were intragastrically administered the powdered hot-melt extrudate (approximately 100~200 um) and powdered tablets (approximately 100~200 um) at a dose of 50 mg/ kg of cilostazol, after sonication (to remove air bubbles) for 30 s in 10 mL injectable water. Cilostazol was administered after a 6 h fasting; feeding was resumed 4 h after drug administration. Blood samples were collected at 8-time-points, at 0, 1, 2, 3, 6, 9, 12, and 24 h after administering the drug as per reference [25]. This animal experiment was performed following the protocol review and approval by the Animal Experimental Ethics Committee of Korea Preclinical Center (P173018, 6 June 2017).

## 3. Results

### 3.1. Processing Temperature of Hot-Melt Extrusion (HME)

The first hot-melt extrudate formulation was prepared using the hot-melt granulation process as previously described by Chen et al. [22]. Our formulation used 10% cilostazol: 45% Kollidon VA 64: 45% Avicel^®^ and screw speed, and processing temperature were 60 rpm, and 170 °C, respectively. The results, including visual observations, indicated that cilostazol was unevenly distributed, with a content uniformity of >15% relative standard deviation (RSD). Notably, HME requires high-temperature and high-energy during the production process, and thus, can induce mixing at the molecular level [7,8,9,10]. Therefore, the use of Avicel^®^, which is not a thermoplastic excipient in the formulation, could not produce a solid dispersion with homogeneous and solubility increased.

The second formulation was prepared using 40% cilostazol: 20% Kollidon VA64: 40% Soluplus^®^, this showed high homogeneity (<2% RSD) based on the content uniformity test. Soluplus^®^, a thermoplastic polymer with excellent solubility, was used in this formulation. From this formulation, a high drug loading was possible because thermoplastic polymers were used. Based on the first granulation method, a second formulation could be built. The differences of manufacturing process between the first and the second formulation were forced to change the processing temperature and screw speed due to the difference between the granulation and the extrusion method. To maximize the solubilization of cilostazol, the production was carried out under various temperature conditions (120–200 °C), and the results are presented in Figure 1. The dissolution results demonstrated that cilostazol exhibited the most enhanced drug release behavior at a processing temperature (HME) of 120 °C. As the processing temperature increased to 160 and 200 °C, lower drug release patterns were observed. This appeared to be due to the relatively low glass transition temperatures of Kollidon^®^ and Soluplus^®^, indicating that when HME occurred in the presence of Kollidon^®^ it increased the stability of the drug through the lowering of the process temperature. Furthermore, the hot-melt extrudates cannot be prepared at a processing temperature of below 100 °C. We observed that a processing temperature of 120 °C was optimum for producing the hot-melt extrudates, with the maximum solubilizing effect and the lowest temperature. In general, drug substances tend to be unstable to heat, and a low processing temperature is desirable during the HME technology [26].

### 3.2. The Effect of Surfactant Concentration on Dissolution Media

To evaluate the solubilization effect of the hot-melt extrudate in the second formulation, the drug release rates of the API powder and hot-melt extrudate were compared. In addition, to confirm the surfactant effect during dissolution, the dissolution tests were carried out using various surfactant concentrations (Figure 2). As shown in Figure 2A, in the medium containing 0.5% SLS (in accordance with USP method), the API powder, which has no solubilization effect, demonstrated almost 100% drug release in 2 h; the hot-melt extrudate with a solubilization effect showed a dissolution rate of 80% in 2 h. HME technology has been effectively used to increase the solubility and dissolution rate of poorly water-soluble drugs [26]. Although variations are dependent on drug characteristics, it is uncommon for a poorly soluble API powder to demonstrate a higher dissolution rate than the HME product. This phenomenon was likely due to the high surfactant concentration; further dissolution tests were carried out as the concentration of the surfactant in the dissolution media was lowered to 0.3%, 0.1%, and 0% (Figure 2B–D). The results revealed that, compared to the hot-melt extruded product (dotted line), the dissolution rate of the API powder (solid line) decreased drastically as the surfactant ratio in dissolution media was reduced. Compared to the API powder, the hot-melt extrudate reported a slight decrease in the dissolution rate as the surfactant concentration in the medium decreased. Thus, the dissolution rate trends of the API powder and HME reversed between 0.3% and 0.1%. In 0.3% SLS medium, the dissolution rate of the API powder was higher than that of the HME product; in 0.1% SLS medium, the dissolution rate of the API powder was lower. Based on the dissolution tests, the solubilization effect was observed at a lower concentration of the SLS medium when HME formulation was produced. However, it was difficult to establish a strategy for determining a suitable pharmaceutical approach solely by solubilizing effect without evaluating absorption.

Thus, parameter sensitivity analysis (PSA) was conducted to assume the factors affecting absorption. In the previous in vitro step, cilostazol was found to be a drug that yielded limited information during dissolution test, because of the large variation in results. We thus concluded that it would not be appropriate to evaluate solubilization solely using a dissolution test to confirm the solubilization effect. To solve this problem, a formulation strategy was established using PSA-based computational simulation. This PSA could be used to assess the factors with the greatest effect on the bioavailability of cilostazol, such as maximum concentration (C_max_) and area under the curve (AUC). Patel and Rajput, using nano-spray and solid dispersion systems as solubilization methods [27], selected the factors expected to affect the bioavailability of cilostazol, and the range of comparability of the factors was determined to the farthest extent possible [27]. Figure 3 shows the PSA results of the drug concentration, particle density, particle size, permeability, and precipitation temperature of the bile salt.

The PSA results indicated that the AUC increased as the particle density decreased and the concentration of the bile salt increased, and the bile salt demonstrated the greatest effect as the AUC increased. Notably, a 10-fold increase in particle density induced approximately a 50% decrease in the AUC, consistent with Jinno et al., [28], who reported a decrease in bioavailability depending on the particle density. Thus, the PSA results accurately predicted these actual results [28]. Therefore, as shown in Figure 3, we focused on the effect of bile salt concentration on the bioavailability of cilostazol. Furthermore, we need to establish a method to induce bile salt formation. Thus, addition of lipids to the formulation, to induce bile salts in the intestine, is expected to reduce the degree of individual differences and enhance bioavailability [27,29].

### 3.3. Optimized Hot-Melt Extruded Formulation Based on PBPK Simulation (Third Formulation)

Based on in vitro tests, it is impossible to measure and predict the effect of inducing the release of bile salts into the small intestine. However, the induction or presence of bile salts has a profound effect on the bioavailability of cilostazol (Figure 3). The addition of lipids to the formulation, to induce bile salts in the intestine, is expected to reduce the degree of individual differences and enhance bioavailability [27,29]. In this study, a lipid-based formulation was prepared using the HME technology based on the results of the PBPK simulation. HME encapsulated the API with minimum lipid amounts evenly distributed under optimized temperature and pressure, allowing the surfactant induction within a short time. Consequently, we prepared a hot-melt extrudate with an even distribution of the vitamin E derivative, TPGS^®^, so that the solid-state extrudate could be maintained at room temperature and produced a lipid-enclosed formulation with a small particle size. Finally, the extrudates were frozen and crushed.

Figure 4 presents the dissolution rate curve of the optimized formulation prepared with a lipid-base, which does not significantly differ from the hot-melt extrudate curve prepared using the hydrophilic polymer (2nd formulation) (Figure 2). In the in vitro test, the optimized hot-melt extruded formulation (PBPK simulation-based) did not considerably differ from 2nd hot-melt extruded formulation; however, the PBPK simulation demonstrated that cilostazol was highly sensitive to surfactant-induced factors. Thus, the improvements introduced in the optimized formulation were expected to increase the rate of absorption by promoting the induction of bile salts in vivo [30].

### 3.4. Physicochemical Properties of the Third Formulation

Raman spectroscopy, XRD, and SEM measurements were performed to determine the changes in cilostazol during the HME process. These results are shown in Figure 5, Figure 6 and Figure 7, respectively. Raman spectroscopy (Figure 5), which focuses on the particle surface, demonstrated cilostazol peaks only in the cilostazol powder (A) and the physical mixture (H) without the HME process. On the other hand, cilostazol was not observed on the surface of all the extrudates when the hot-melt extrudate was produced by altering the processing temperature from 120 to 220 °C. Ideally, the lipid matrix uniformly wraps around the particle surface [31]. For better stability, the lowest and optimized temperature of 120 °C was set as the manufacturing temperature [26]. Furthermore, hot-melt extrudates were not produced at temperatures below 120 °C. The XRD data (Figure 6) indicated that the cilostazol crystallinity was maintained in all cilostazol powders, physical mixtures, and hot-melt extrudates. However, the SEM images demonstrated that the pure cilostazol (Figure 7A) and hot-melt extrudate (Figure 7B) were distinct. After the HME process, all the angular crystal structures of cilostazol were well mixed with the lipid. In other words, the cilostazol crystallinity was maintained even after the hot-melt extrusion process and cilostazol was made to clusters with lipid. Thus, it is possible that this formulation maintained the stability of the drug [32,33,34] and may induce adequate bile secretion in the small intestine.

### 3.5. In Vivo Study

Based on the in vivo results of Patel and Rajput’s studies [27], the marketed product showed approximately a 1.5-higher C_max_ and AUC than pure cilostazol. Thus, the in vivo experiment consisted of six rats, three of which were administered the commercially available cilostazol product instead of pure cilostazol, Pletaal^®^ (Otsuka, Japan), and the other three were administered the optimized hot-melt extruded (PBPK simulation-based) formulation. The results of the PK profiles and values are shown in Figure 8 (the average and Table 3. Pharmacokinetic parameters of the reference Pletaal^®^ were 1348.11 ± 776.77 ng/mL (C_max_), 9.0 ± 3.0 h (T_max_), and 13629.60 ± 9997.50 ng·h/mL (AUC_0–24_). For the optimized hot-melt extruded formulation, the values were 1646.87 ± 747.42 ng/mL (C_max_), 1.7 ± 0.6 h (T_max_), and 9476.54 ± 1971.07 ng·h/mL (AUC_0–24_). The optimized hot-melt extruded (PBPK simulation-based) formulation exhibited extremely fast drug absorption kinetics compared to Pletaal^®^.

This indicated that the lipid-based formulation induced bile salts in the intestine after administration, resulting in rapid drug absorption. The earlier PSA results have proven that the absorption increased as the particle density decreased and the concentration of the bile salts increased, with the bile salts demonstrating the greatest effect on the absorption rate. Furthermore, the RSD of AUC in the optimized hot-melt extruded (PBPK simulation-based) formulation was remarkably reduced compared to Pletaal^®^ (Table 3).

We produced a small particle size with an even distribution of lipid via hot melt extrusion technology that could induce the surfactant rapidly and constantly. The in vivo results of the computational simulation-based formulation showed that its dissolution rate was not improved compared to that of the original formulation. However, these test results also showed that the rate of absorption of the optimized formulation was higher than that of the conventional formulation.

In addition, since the animals were fed in 4 h after drug administration, the C_max_ was increased at 6 h. Furthermore, the difference in plasma concentrations between the 3 h and 6 h samples was considerably large in the Pletaal^®^ group (control), indicating that the surfactant was induced by feeding. In other words, if feeding was restricted, then a significantly lower AUC value with Pletaal^®^ could be expected. In contrast with the optimized hot-melt extruded (PBPK simulation-based) formulation, the C_max_ occurred rapidly irrespective of feeding, and the absorption was not increased after meals.

Pletaal^®^ treatment can be problematic as individual PK differences were considerably large between the three rats. This large deviation could lead to adverse effects and challenges in administering the appropriate cilostazol dose. In the optimized hot-melt extruded (PBPK simulation-based) formulation, C_max_ occurred at approximately the same time and no individual differences more than two and half times were observed. An individual difference occurring around twice the value was considered in the acceptable range based on previous reports indicating that gastrointestinal surfactant concentrations differ by as much as five times with the same high-fat diet [27]. In vitro results of the optimized hot-melt extruded (PBPK simulation-based) formulation showed that the dissolution rate was not improved compared to that of the original formulation (Figure 4); however, this in vivo result demonstrated that the rate of absorption of the optimized formulation was higher with lower deviation than that of the marketed formulation. Furthermore, administration was regardless of meal consumption.

## 4. Conclusions

This research indicates that PBPK simulation may complement in vitro and in vivo investigations, increasing the success rate of formulation studies. The PSA of the PBPK simulation was used to develop an optimized hot-melt extruded formulation. The PSA indicated that surfactant induction is a key point in cilostazol formulation; thus, we produced a small particle size with an even lipid distribution that could induce the surfactant. As a result, the rate of absorption of the optimized formulation was higher than that of the marketed drug. Currently, there is a limit to the integration of PBPK simulations; nevertheless, this method has been actively used in pharmaceutics, especially in formulation strategies.

## Figures and Tables

**Figure 1 pharmaceutics-12-00757-f001:**
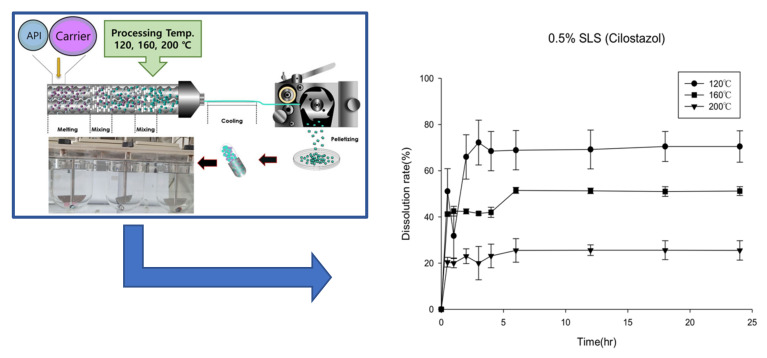
Dissolution profiles of hot melt extruded cilostazol (2nd formulation) at various processing temperatures (120, 140 and 160 °C, *n* = 3).

**Figure 2 pharmaceutics-12-00757-f002:**
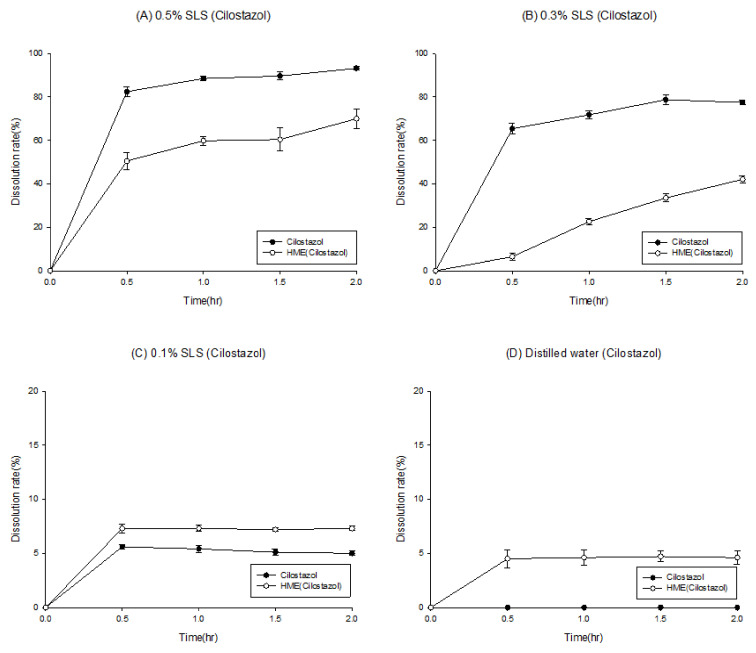
Results of dissolution rate of cilostazol powder (solid line) and hot-melt extrudate (2nd formulation; dotted line) on various sodium lauryl sulfate (SLS) concentrations (Y-axis is not fixed to clearly show the difference in dissolution rate).

**Figure 3 pharmaceutics-12-00757-f003:**
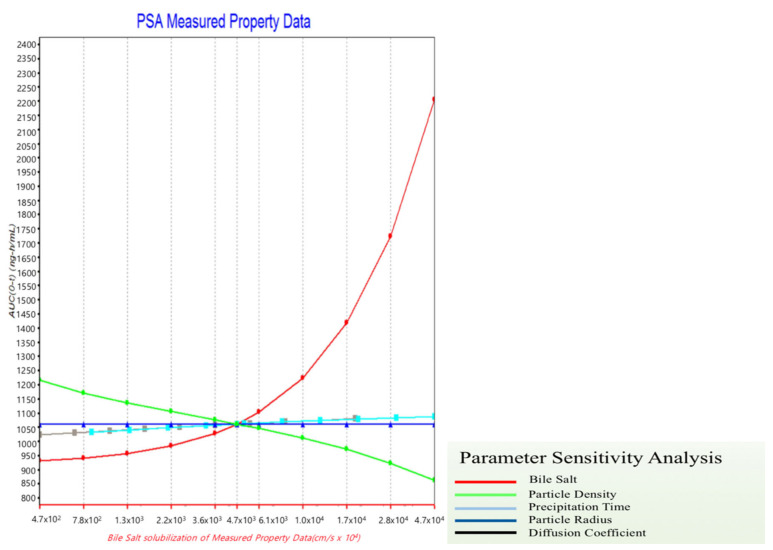
The factors affecting the area under curve (AUC) based on parameter-sensitive analysis (PSA) results.

**Figure 4 pharmaceutics-12-00757-f004:**
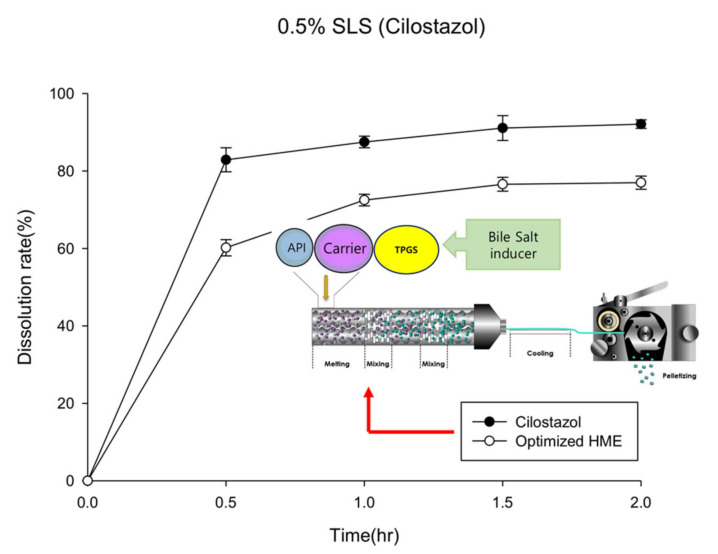
Dissolution profiles of the optimized hot-melt extruded formulation (computational simulation-based) in water containing 0.5% sodium lauryl sulfate (SLS, *n* = 3).

**Figure 5 pharmaceutics-12-00757-f005:**
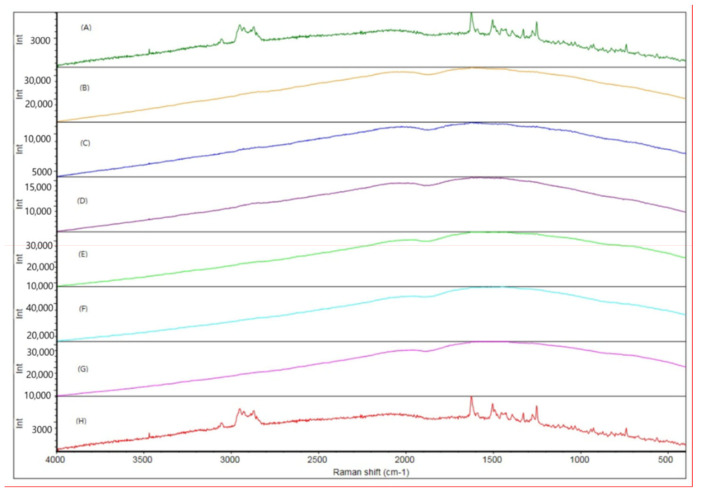
Raman spectroscopy: (**A**) cilostazol, hot-melt extrudate prepared at (**B**) 120, (**C**) 140, (**D**) 160, (**E**) 180, (**F**) 200 and (**G**) 220 °C, (**H**) physical mixture.

**Figure 6 pharmaceutics-12-00757-f006:**
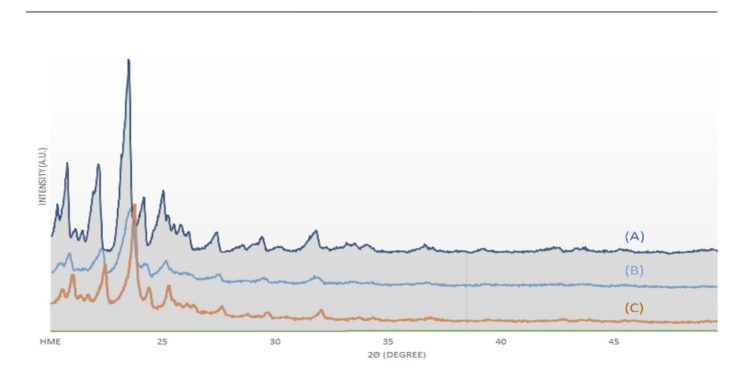
X-ray diffraction (XRD) images: (**A**) cilostazol, (**B**) physical mixture, and (**C**) optimized hot-melt extrudate.

**Figure 7 pharmaceutics-12-00757-f007:**
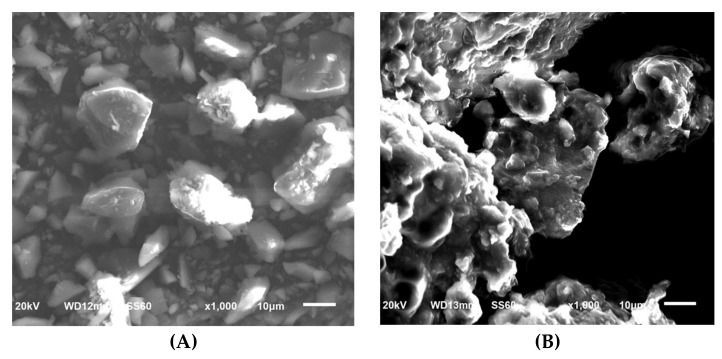
Scanning electron microscopy (SEM) images: cilostazol (**A**), optimized hot-melt extrudate (**B**).

**Figure 8 pharmaceutics-12-00757-f008:**
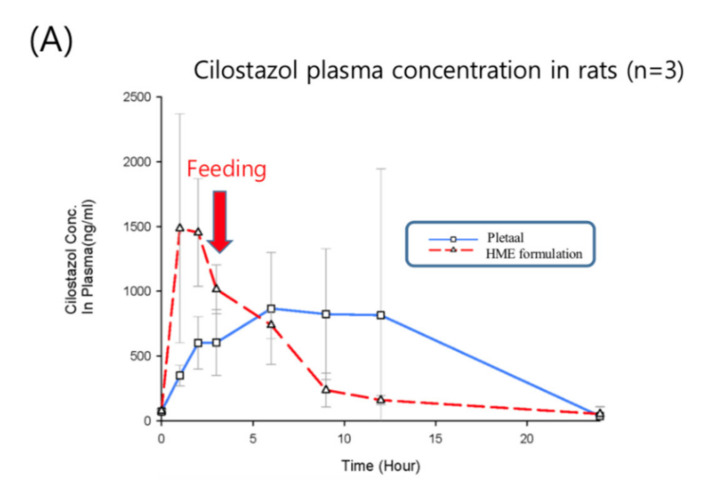
Plasma concentration time curves of Cilostazol after oral administration of Pletaal^®^ and hot melt extrusion (HME) formulation to a rat (n = 3); mean value with deviation (**A**); individual value of Pletaal^®^ (**B**) and HME formulation (**C**) in each rat.

**Table 1 pharmaceutics-12-00757-t001:** Composition of Formulations and Processing Conditions.

Parameters	First Formulation	Second Formulation	Third Formulation *
Cilostazol	10%	40%	40%
Avicel	45%		
Kollidon	45%	20%	20%
Soluplus	-	40%	
TPGS	-		32%
Cremophor	-		8%
Processing temperature	170 ℃	120 ℃	120 ℃
Screw Speed	60 rpm	100 rpm	100 rpm

* Computational simulation-based formulation.

**Table 2 pharmaceutics-12-00757-t002:** Physiological Factors of 300 g Rat Provided by Gastroplus™.

Tissue	Volume	Tissue–Plasma Ratio	Fut/FuExt
**Lung**	2.52	2.26	0.016
**Arterial supply**	6.72	0	0
**Venous return**	13.56	0	0
**Adipose**	12	7.04	0.001
**Muscle**	146.4	1.09	0.037
**Liver**	12.36	1.91	0.015
**ACAT gut**	0	0	0
**Spleen**	0.72	1.04	0.023
**Heart**	1.44	1.55	0.025
**Brain**	1.48	4.42	0.007
**Kidney**	4.44	1.78	0.014
**Skin**	48	2.56	0.025
**Reproductive organ**	3	1.79	0.014
**Red marrow**	2.24	2.36	0.032
**Yellow marrow**	4.98	7.04	0.001
**Rest of the body**	29.31	1.06	0.023

**Table 3 pharmaceutics-12-00757-t003:** Pharmacokinetic parameters of cilostazol following oral administration of Pletaal^®^ and optimized hot-melt extrudates (computational simulation-based formulation) (*n* = 3).

	Pletaal^®^			Optimized Hot-Melt Extrudates		
	C_max_ (ng/mL)	T_max_ (h)	AUC (ng·h/mL)	C_max_ (ng/mL)	T_max_ (h)	AUC (ng·h/mL)
1	2120.22	12.0	24856.09	2507.32	1.0	11714.83
2	1357.35	6.0	10345.90	1158.75	2.0	8714.67
3	566.76	9.0	5687.10	1274.54	2.0	8000.12
Mean ± SD	1348.11 ± 776.77* (57.62)	9.0 ± 3.0* (33.3)	13629.69 ± 9997.50 * (73.35)	1646.87 ± 747.42* (45.38)	1.7 ± 0.6* (34.6)	9476.54 ± 1971.07* (20.80)

* The number in parenthesis indicates relative standard deviation (RSD).

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
