# Peer review of "Preparation of Hot-Melt Extruded Dosage Form for Enhancing Drugs Absorption Based on Computational Simulation"

_pharmaceutics, 2020, doi:10.3390/pharmaceutics12080757_

Round 1

Reviewer 1 Report

The present manuscript aims to formulate a hot-melt extruded dosage form for enhancing the bioavailability of cilostazol, independently by food intake. The overall study can be interesting but some points should be addressed.

Main points:

a) Both in the abstract and the introdution it is claimed that a Hot-melt extruded dosage form of cilastazol has been formulated to "enhance" or "control" the solubility. However, no solubility studies were performed.

b) it is not clear the rationale about the choice of the First and second formulation (Table 1). How have been selected? Please provide more details.

c) The parameter-sensitive analysis (PSA) applied to the develepment of the final formulation is a crucial point of the manuscript. But it has not the right relevance in the text. How this methodology guided the optimization of the formulation through a computational-simulation approach? Motreover, the results relative to PSA (line 189-196) and Figure 3 are little confusing. Please improve them.

d) I think that only one experiment conducted on six rats (three for the experimental group and three for the control) and one condition is very poor to state in the title and the conclusion that "the formulation enhances drug absorption regardless food intake" or "it can be administered regardless of meal consuption". Moreover, it is not clear why the optimazed formulation show a pharmacokinetic profile much less influenced by food intake and which is the relationship between this result and the in vitro simulation by PSA. Please discuss more about that.

Other points:

Which is the size of the solid extrudates tested as "Second Fomulation" and "third formulation" both for in vitro and in vivo studies? The size can markedly affect the release results and/or the pharmacokinetic profile.

How the content uniformity in First formulation and Second Formulation was determined? (Line 145 and Line 150). Please add this information.

It is not clear that results in Figure 1 and Figure 2 refer to "Second formulation"

For what reason the in vitro release study was performed using pure poweder cilostazol, while the in vivo study using the commercial product containing cilostazol "Pletaal". Which dosage form is Pletaal?

Author Response

Response to Reviewers’ Comments

We, the authors, would like to take this opportunity to express our sincere appreciation to Reviewer 1 and Reviewer 2 for their considerable constructive critiques of our manuscript. The authors have addressed the Reviewers’ concerns below in a point-by-point fashion and detailed the edits to the body of the manuscript, references, and Figures. The changes made to the manuscript are shown in red or bold using track change; these changes, which are based on the Reviewers’ comments, have improved the strength of the manuscript.

Reviewer 1

The present manuscript aims to formulate a hot-melt extruded dosage form for enhancing the bioavailability of cilostazol, independently by food intake. The overall study can be interesting,  but some points should be addressed.

Main points:

  1. a) Both in the abstract and the introdution it is claimed that a Hot-melt extruded dosage form of cilastazol has been formulated to "enhance" or "control" the solubility. However, no solubility studies were performed.

☞ The authors changed the “solubility” to “dissolution rate” in Abstract and Introduction part in line 11,12 38,42, 57, 60.  

  1. b) it is not clear the rationale about the choice of the First and second formulation (Table 1). How have been selected? Please provide more details.

☞ The first formulation was prepared using the hot-melt Granulation process as previously described by Chen et al. [22]. The first formulation, including visual observations, indicated that cilostazol was unevenly distributed, with a content uniformity of > 15% relative standard deviation (RSD). However; the second formulation utilized with hot melt extrusion technology showed high homogeneity (< 2 % RSD) based on the content uniformity test.

  1. c) The parameter-sensitive analysis (PSA) applied to the develepment of the final formulation is a crucial point of the manuscript. But it has not the right relevance in the text. How this methodology guided the optimization of the formulation through a computational-simulation approach? Motreover, the results relative to PSA (line 189-196) and Figure 3 are little confusing. Please improve them.

☞ Thank you for very nice comments. The authors tried to improve the explanation of PSA results and Figure 3m line 219~229.

“ In the previous in vitro step, cilostazol was found to be a drug that yielded limited information during dissolution test, because of the large variation in results. We thus concluded that it would not be appropriate to evaluate solubilization solely using a dissolution test to confirm the solubilization effect. To solve this problem, a formulation strategy was established using PSA-based computational simulation. This PSA could be used to assess the factors with the greatest effect on the bioavailability of cilostazol, such as maximum concentration (Cmax) and area under the curve (AUC). Patel and Rajput, using nano-spray and solid dispersion systems as solubilization methods [27], selected the factors expected to affect the bioavailability of cilostazol, and the range of comparability of the factors was determined to the farthest extent possible [27]. Figure 3 shows the PSA results of the drug concentration, particle density, particle size, permeability, and precipitation temperature of the bile salt.”

  1. d) I think that only one experiment conducted on six rats (three for the experimental group and three for the control) and one condition is very poor to state in the title and the conclusion that "the formulation enhances drug absorption regardless food intake" or "it can be administered regardless of meal consuption". Moreover, it is not clear why the optimazed formulation show a pharmacokinetic profile much less influenced by food intake and which is the relationship between this result and the in vitro simulation by PSA. Please discuss more about that.

☞ Thank you for very insight comment. The authors changed the title to “Preparation of Hot-Melt Extruded Dosage Form for Enhancing Drugs Absorption Based on Computational Simulation.” and the sentence was removed “Furthermore, it can be administered regardless of meal consumption” in conclusion in accordance with reviewer’s comment. Furthermore, we added a paragraph in line 314~318, “we produced a small particle size with an even distribution of lipid via hot melt extrusion technology that could induce the surfactant rapidly and constantly. The in vivo results of the computational simulation-based formulation showed that its dissolution rate was not improved compared to that of the original formulation. However, these test results also showed that the rate of absorption of the optimized formulation was higher than that of the conventional formulation.” Thank you again.

Other points:

  1. Which is the size of the solid extrudates tested as "Second Fomulation" and "third formulation" both for in vitro and in vivo studies? The size can markedly affect the release results and/or the pharmacokinetic profile.

☞ The granule or extrudate with a size of 100~200um were selected and applied to all experiments. We added this sentence in line 96~97.

  1. How the content uniformity in First formulation and Second Formulation was determined? (Line 145 and Line 150). Please add this information.

☞ Drug release and content uniformity section (2.3) were updated and added in line 102~125.

  1. It is not clear that results in Figure 1 and Figure 2 refer to "Second formulation".

☞ The authors revised the legend of Fig 1 and 2. 2nd formulation was used for Fig 1 and 2. Thank you for comment.

  1. For what reason the in vitro release study was performed using pure poweder cilostazol, while the in vivo study using the commercial product containing cilostazol "Pletaal". Which dosage form is Pletaal?

☞ The authors thought that it was more meaningful to compare with a commercial product than cilostazol powder. This is because the patients were administered cilostazol with pharmaceutical dosage form but API alone. If in vivo experiments were conducted with cilostazol powder, we believe that too general results have been obtained. Furthermore, many manuscripts showed that the comparison with cilostazol is already performed. When comparing cilostazol powder with formulation in all previous studies, it was confirmed that the formulation was superior in dissolution. However, the authors also thought that it would have been better if we had compared API, commercial product, and hot melt extrudate together.

Reviewer 2 Report

The work is interesting and the results seem consistent even if information on the methods and the some parts of the discussion should be implemented

In particular, the solid state has been characterized with different techniques but the discussion is lacking with respect to the data obtained. The authors must delve into this part of the discussion. The technological approach adopted is interesting but the physical change of the active formulation is the basis of the result obtained, therefore it is absolutely necessary to implement the discussion on the results obtained from the analysis of the solid state otherwise it makes no sense to present them.

 I believe the work is suitable for publication in pharmaceutics but needs major revisions before being accepted.

Acronyms must be developed the first time they appear in the text, verify starting from the abstract

In the introduction section authors should clirify what they mean for  “have a unique post-absorptive factor”

Materials section: all the commercial names of the used excipients have to be developed with the common nomenclature indicated in brackets

Line 82: temperature of 170? Which degree? Celsius? Please add the information

Line 83: drug-loading was of 40%....but I do not understand the percentages of the other components as a percentage as they have been compared to the variation in the percentage of drug loaded. Please clarify.  Authors have reported a table with the formulation composition. The same information can be removed from the text in method section and in part discussed in results section.

Why have experimental formulations and conditions been changed at the same time? in order to have a correct comparison between the results, the formulations should be processed in the same conditions in order to correctly evaluate the influence of the composition or process parameters on the final result. In this way the results are not correctly comparable. The authors should add the results from comparable experiments (same formulation and different process parameters or different formulations but same process parameters) and then optimize based on the results.

Par.2.3: Refer correctly to the USP considered in the text and to the specific test used

Specify how the result will be expressed

Authors have to add the calibration curve and the analytical method to identify the drug by HPLC. The information in the text are not enough. Add the regression equation. Was the maximum absorption length determined experimentally by the authors or referred to literature data? in both cases, information must be entered.

SEM information is not enough, you need to enter the frame, working distance and magnification used.

Line 141. Please remove the sentence…we are unable…..or rephrase.

Line 145: authors declare a RSD >15% but in the methods section it is not explained how it is determined

The text inside the figure 1 on the left is difficult to read. The figure must be implemented. Add degrees celsius to the chart legend.

The authors should be careful to talk about solubility and the amount of dissolution as they are two absolutely different parameters. The use of the surfactant improves solubility as the dissolution medium is no longer just water therefore the physical behavior changes; the speed and quantity dissolved over time of the asset is very different and can depend on various factors. Has the sink condition been assessed? the authors don't mention it. In an in vitro study with USPII (closed system), more attention should be given to the experimental parameters that make the results correct and solid. Probably in a study of this type, in order not to have the negative influence of saturation on the behavior of the drug in dissolution, an experiment with a USP IV (flow trough) had to be programmed.

Figures have been implemented in risolution.

Line 232, 235: please add degrre celsius

Figure 7. SEM images are not particularly sharp. They should be acquired with greater contrast. Furthermore, judging by the intense light of some points, the sample was not adequately metallized. In the caption change left and right with a and b in brackets and addi t to the figure.

Line 240. Particles??? I don’t see particles. I see a cluster state but no particles. Please revise the discussion according to the results obtained.

The chemical- physical stability could be assessed. Which was the encapsulation efficiency?  How was the dose to be administered in vivo calculated? based on initial drug-loading? this procedure is not correct if you do not have the data relating to the encapsulation efficiency of the system.

The discussion and the correlation between data on solid state has been revised.

How does solid state affect bioavailability? is it just the combination with surfactants? or does physical interaction between the components play a fundamental role? on the other hand this is the purpose of technological intervention in general. The authors need to highlight it better, even in the final part of the work.

Author Response

We, the authors, would like to take this opportunity to express our sincere appreciation to Reviewer 1 and Reviewer 2 for their considerable constructive critiques of our manuscript. The authors have addressed the Reviewers’ concerns below in a point-by-point fashion and detailed the edits to the body of the manuscript, references, and Figures. The changes made to the manuscript are shown in red or bold using track change; these changes, which are based on the Reviewers’ comments, have improved the strength of the manuscript.

Reviewer 2

  1. The work is interesting and the results seem consistent even if information on the methods and the some parts of the discussion should be implemented. In particular, the solid state has been characterized with different techniques but the discussion is lacking with respect to the data obtained. The authors must delve into this part of the discussion. The technological approach adopted is interesting but the physical change of the active formulation is the basis of the result obtained, therefore it is absolutely necessary to implement the discussion on the results obtained from the analysis of the solid state otherwise it makes no sense to present them.  I believe the work is suitable for publication in pharmaceutics but needs major revisions before being accepted.

☞ We, the authors, would like to take this opportunity to express our sincere appreciation to Reviewer 2 for his/her considerable constructive critiques of our manuscript. The authors have addressed the Reviewer’s concerns below in a point-by-point fashion and detailed the edits to the body of the manuscript, references, and Figures. The changes made to the manuscript are shown in red or bold using track change; these changes, which are based on the Reviewer’s comments, have improved the strength of the manuscript.

  1. Acronyms must be developed the first time they appear in the text, verify starting from the abstract

☞ The authors changed in line 15th in abstract from “PBPK” to “physiologically based pharmacokinetic (PBPK)”, thank you for comment.

  1. In the introduction section authors should clirify what they mean for  “have a unique post-absorptive factor”

☞ The authors changed in line 43th in Introduction from “post-absorptive factor” to “absorption factor”, sorry for making you confuse.

  1. Materials section: all the commercial names of the used excipients have to be developed with the common nomenclature indicated in brackets

☞ In accordance with reviewer’s comments, the authors revised in 2.1 Material section in line 67~71. “Vinylpyrrolidone-vinyl acetate copolymer(Kollidon® VA64), polyvinyl caprolactam–polyvinyl acetate–polyethylene glycol graft copolymer(Soluplus®), D-α-Tocopheryl polyethylene glycol 1000 succinate (TPGS®), and Polyethoxylated castor oil (Cremophor®) were kindly donated by BASF (Ludwigshafen, Germany); and microcrystalline cellulose (Avicel®) was provided by FMC”

  1. Line 82: temperature of 170? Which degree? Celsius? Please add the information

☞ We added the “℃” after 170. The authors feel sorry for that. 

  1. Line 83: drug-loading was of 40%....but I do not understand the percentages of the other components as a percentage as they have been compared to the variation in the percentage of drug loaded. Please clarify.  Authors have reported a table with the formulation composition. The same information can be removed from the text in method section and in part discussed in results section.

☞ The authors revised this “2.2. preparation of hot-melt extrudates” section. We tried to explain the differences between the formulations and removed the same explanation from the table and the text. Thank you for very insightful comment.

  1. Why have experimental formulations and conditions been changed at the same time? in order to have a correct comparison between the results, the formulations should be processed in the same conditions in order to correctly evaluate the influence of the composition or process parameters on the final result. In this way the results are not correctly comparable. The authors should add the results from comparable experiments (same formulation and different process parameters or different formulations but same process parameters) and then optimize based on the results.

☞ In this paper, comparing the second formulation (hot melt extrusion) to the third formulation (hot melt extrusion and computational simulation) is the main theme. We would say that the first formulation was a preliminary study of cilostazol using a hot melt GRANULATION method followed by Chen [22], by using excipient with none-thermoplastic property such as Avicel to obtain simple granules, Not extrudate. Based on the first granulation method, a second formulation could be built, and finally, a 3rd formulation could be completed through computational simulation. Therefore, the differences between the first, and the second and third formulations were forced to change the manufacturing temperature conditions due to the difference between the granulation and the extrusion method. However, in the second and third formulations, the formulation was changed, but the manufacturing processing conditions were NOT. The authors revised 2.2 section. Thank you for very insightful comment

  1. 2.3: Refer correctly to the USP considered in the text and to the specific test used, Specify how the result will be expressed

☞ The authors added revised 2.3 section in accordance with the reviewer’s comment and added the dissolution calculation equation in line 112~114.

  1. Authors have to add the calibration curve and the analytical method to identify the drug by HPLC. The information in the text are not enough. Add the regression equation. Was the maximum absorption length determined experimentally by the authors or referred to literature data? in both cases, information must be entered.

☞ The authors added calibration curve in supplement data and mentioned the calibration equation and R2 value in 2.3 Section. We had a mistake for wavelength in original manuscript. The wavelength was 257 nm in accordance with USP 42-NF37 “Cilostazol Tablets”  

  1. SEM information is not enough, you need to enter the frame, working distance and magnification used.

☞ The authors updated SEM information in 2.5 Section in line 149. “SEM was used to detect any physical changes in the matrix systems. The images were collected using a JSM-6510 scanning electron microscope (JEOL Ltd., Tokyo, Japan) at 20 kV and magnification of 500

  1. Line 141. Please remove the sentence…we are unable…..or rephrase.

☞ We removed this sentence. Thank you for very helpful comment.

  1. Line 145: authors declare a RSD >15% but in the methods section it is not explained how it is determined

☞ The author added the preparation method for evaluating the content uniformity of cilostazol in 2.3 section.

  1. The text inside the figure 1 on the left is difficult to read. The figure must be implemented. Add degrees celsius to the chart legend.

☞ The authors updated the Figure 1, text inside figure and legend.

  1. The authors should be careful to talk about solubility and the amount of dissolution as they are two absolutely different parameters. The use of the surfactant improves solubility as the dissolution medium is no longer just water therefore the physical behavior changes; the speed and quantity dissolved over time of the asset is very different and can depend on various factors. Has the sink condition been assessed? the authors don't mention it. In an in vitro study with USPII (closed system), more attention should be given to the experimental parameters that make the results correct and solid. Probably in a study of this type, in order not to have the negative influence of saturation on the behavior of the drug in dissolution, an experiment with a USP IV (flow trough) had to be programmed.

☞ We evaluated drug release in dissolution media containing various concentrations of SLS from 0, 0.1, 0.3 and 0.5% (Figure 2) and explained about this on 3.2 section. In accordance with USP-NF “Cilostazol Tablet”, USP suggests deionized water with 0.3% SLS as dissolution media. However, Nam et al.(Double controlled release of highly insoluble cilostazol using surfactant- driven pH dependent and pH-independent polymeric blends and in vivo bioavailability in beagle dogs, International Journal of Pharmaceutics, 2019, 558, 284-290) suggests that the sink condition of cilostazol is deionized water with 0.5% SLS. Based on our research (Fig.2), it would not be appropriate to evaluate solubilization solely using a dissolution test to confirm the solubilization effect. To solve this problem, a formulation strategy was established using PSA-based computational simulation. This PSA could be used to assess the factors with the greatest effect on the bioavailability of cilostazol, such as maximum concentration (Cmax) and area under the curve (AUC). It was difficult to establish a strategy for determining a suitable pharmaceutical approach solely by solubilizing effect without evaluating absorption. Thank you for very insightful comment, and we added sentences in line 219~229.

  1. Figures have been implemented in risolution.

☞ The authors increased resolutions to be clear in Fig 4,5,6 and 8. 

  1. Line 232, 235: please add degrre Celsius

☞ We added ℃ in line 275 and 277 and 278. We feel sorry for that.

  1. Figure 7. SEM images are not particularly sharp. They should be acquired with greater contrast. Furthermore, judging by the intense light of some points, the sample was not adequately metallized. In the caption change left and right with a and b in brackets and addi t to the figure.

☞ The authors thought the reviewer has made the correct point. However, although the SEM image was not sharp, it seems that a lipid substance surrounds cilostazol in the case of figure 7-(b). And we added (a) and (b) instead of (left) and (right).

  1. Line 240. Particles??? I don’t see particles. I see a cluster state but no particles. Please revise the discussion according to the results obtained.

☞ The authors changed “powdered cilostazol” to “pure cilostazol: in line 280.

  1. The chemical- physical stability could be assessed. Which was the encapsulation efficiency?  How was the dose to be administered in vivo calculated? based on initial drug-loading? this procedure is not correct if you do not have the data relating to the encapsulation efficiency of the system.

☞The amount of cilostazol encapsulated with lipid material during the HME process can be 100%. This was proven in the content test. No degradation occurred during the HME process. Thus, dose of 50 mg/ kg of cilostazol was administered in Rats. However, we revised the sentence in line 282 to be clear on this. Thank you for comment.

  1. How does solid state affect bioavailability? is it just the combination with surfactants? or does physical interaction between the components play a fundamental role? on the other hand this is the purpose of technological intervention in general. The authors need to highlight it better, even in the final part of the work.

☞The authors added a paragraph in line 314~318 to highlight our outcomes and results.

Round 2

Reviewer 1 Report

The authors have addessed the comments. The manuscript is suitable for pubblication.

Author Response

We appreciate this.

Reviewer 2 Report

The authors made the required changes, and where not possible they well argued the critical issues related to the impossibility of improving the data obtained. These critical issues have been explained and highlighted in the revised text which now, in my opinion, has a greater fluidity and critical sense.

I have just a few other small suggestions before publication that I would like the authors to consider.

Figure 1 in supplementary: please remove the % and leave only the concentration in mg/L

Line 86 to 91: please move to discussion section.

Line 119 please correct in 0.45 -1 µm, also adding the supplier

Line 123 please remove the percentage, leave only the concentration in mg/L

Line 313: Start the sentence with a capital letter

Pay attention: 

  1. Figure 7. SEM images are not particularly sharp. They should be acquired with greater contrast. Furthermore, judging by the intense light of some points, the sample was not adequately metallized. In the caption change left and right with a and b in brackets and addi t to the figure.

☞ The authors thought the reviewer has made the correct point. However, although the SEM image was not sharp, it seems that a lipid substance surrounds cilostazol in the case of figure 7-(b).

I think the image has absolutely improved, this answer is not acceptable. If the authors are unable to produce a better image due to difficulties in preparing the sample, probably due to the nature of the sample itself, they must explain it in the text, so as to facilitate the work of other researchers who want to study and take inspiration from their results. . This is the meaning of scientific publication, not only the publication of a result, but the highlighting of particular critical issues to facilitate the scientific community in future studies.

  1. Line 240. Particles??? I don’t see particles. I see a cluster state but no particles. Please revise the discussion according to the results obtained.

☞ The authors changed “powdered cilostazol” to “pure cilostazol: in line 280.

I don’t understand the correction. The comment concerns the solid state which is not a particle but a crystalline one. Please revise.

Author Response

Response to Reviewers’ Comments

Reviewer 2

The authors made the required changes, and where not possible they well argued the critical issues related to the impossibility of improving the data obtained. These critical issues have been explained and highlighted in the revised text which now, in my opinion, has a greater fluidity and critical sense. I have just a few other small suggestions before publication that I would like the authors to consider.

Figure 1 in supplementary: please remove the % and leave only the concentration in mg/L

☞ We remove “(%)” and leave only “ug/mL”

Line 86 to 91: please move to discussion section.

☞ We moved these sentences to in line 175~180. Thank you very much.

Line 119 please correct in 0.45 -1 µm, also adding the supplier

☞ In accordance with reviewer’s comment, we revised this and added supplier. We appreciate this.

Line 123 please remove the percentage, leave only the concentration in mg/L

☞ The authors remove the % in this sentence.

Line 313: Start the sentence with a capital letter

☞ We changed it to a capital letter. Sorry for that. 

Pay attention: 

Figure 7. SEM images are not particularly sharp. They should be acquired with greater contrast. Furthermore, judging by the intense light of some points, the sample was not adequately metallized. In the caption change left and right with a and b in brackets and addi t to the figure.

☞ The authors thought the reviewer has made the correct point. However, although the SEM image was not sharp, it seems that a lipid substance surrounds cilostazol in the case of figure 7-(b).

I think the image has absolutely improved, this answer is not acceptable. If the authors are unable to produce a better image due to difficulties in preparing the sample, probably due to the nature of the sample itself, they must explain it in the text, so as to facilitate the work of other researchers who want to study and take inspiration from their results. . This is the meaning of scientific publication, not only the publication of a result, but the highlighting of particular critical issues to facilitate the scientific community in future studies.

☞ We revised the sentences from line 280 to 283.

Line 240. Particles??? I don’t see particles. I see a cluster state but no particles. Please revise the discussion according to the results obtained.

☞ The authors changed “powdered cilostazol” to “pure cilostazol: in line 280.

I don’t understand the correction. The comment concerns the solid state which is not a particle but a crystalline one. Please revise.

☞ We revised the sentences from line 280 to 283. According to reviewer’s concern, we added that “the cilostazol crystallinity was maintained even after the hot-melt extrusion process and cilostazol was made to clusters with lipid.”

We, the authors, would like to take this opportunity to express our sincere appreciation to Reviewer 2 for their considerable constructive critiques of our manuscript.